# Genome-Wide Association Study Reveals the Genetic Basis of Kernel and Cob Moisture Changes in Maize at Physiological Maturity Stage

**DOI:** 10.3390/plants11151989

**Published:** 2022-07-30

**Authors:** Minyan Zhang, Chaoyang Xiangchen, Jiaquan Yan, Yujuan Chengxu, Hao Liu, Chaoying Zou, Guangtang Pan, Yaou Shen, Langlang Ma

**Affiliations:** State Key Laboratory of Crop Gene Exploration and Utilization in Southwest China, Maize Research Institute, Sichuan Agricultural University, Chengdu 611130, China; minyan_z@163.com (M.Z.); xccy498720303@163.com (C.X.); yjq15082436449@163.com (J.Y.); ccxxyyjj57364@163.com (Y.C.); liuhao971229@163.com (H.L.); zoucy2022@163.com (C.Z.); pangt@sicau.edu.cn (G.P.); shenyaou@sicau.edu.cn (Y.S.)

**Keywords:** maize, physiological maturity, moisture content, dehydration rate, genome-wide association study

## Abstract

Low moisture content (MC) and high dehydration rate (DR) at physiological maturity affect grain mechanical harvest, transport, and storage. In this study, we used an association panel composed of 241 maize inbred lines to analyze ear moisture changes at physiological maturity stage. A genome-wide association study revealed nine significant SNPs and 91 candidate genes. One SNP (SYN38588) was repeatedly detected for two traits, and 15 candidate genes were scanned in the linkage disequilibrium regions of this SNP. Of these, genes *Zm00001d020615* and *Zm00001d020623* were individually annotated as a polygalacturonase (PG) and a copper transporter 5.1 (COPT5.1), respectively. Candidate gene association analysis showed that three SNPs located in the exons of *Zm00001d020615* were significantly associated with the dehydration rate, and AATTAA was determined as the superior haplotype. All these findings suggested that *Zm00001d020615* was a key gene affecting moisture changes of maize at the physiological maturity stage. These results have demonstrated the genetic basis of ear moisture changes in maize and indicated a superior haplotype for cultivating maize varieties with low moisture content and high dehydration rates.

## 1. Introduction

Maize (*Zea may* L.) is one of the main food crops worldwide and is an important source of human nutrition, animal feed, and bioenergy. To further improve production efficiency and reduce production costs, mechanized grain harvesting has become a key technology for crop production [1]. However, ears having a high moisture content (MC) at the harvesting stage causes grain mildew and restrict mechanized harvesting [2]. In the natural drying process from physiological maturity to harvest, the initial MC at physiological maturity and the dehydration rate (DR) in the field environment jointly determine the final MC at harvest [3,4]. Therefore, breeding maize hybrids with low MC and high DR at physiological maturity can promote mechanical harvest, improve threshing efficiencies, and reduce additional drying costs [5,6,7].

The MC and DR are quantitative traits mainly controlled by genetic factors [8], although they are affected by changes in the field ambient temperature and humidity [9,10]. Currently, numerous moisture-gain-associated QTL have been detected by linkage analysis. For example, a total of 76 QTL associated with dehydration characteristic parameters were detected using a recombinant inbred line (RIL) population of 208 lines, which explained 1.03% to 15.24% of the phenotypic variation [11]. Liu et al. [12] identified seven QTL related to grain water content using an RIL population, which explained 6.92–24.78% of the phenotypic variation. However, only a few moisture-gain-associated genes were fine-mapped due to the relatively low resolution of linkage mapping [13]. Genome-wide association study (GWAS) is an effective tool for analyzing the genetic structure of complex quantitative traits, which provides a high-resolution strategy for identifying the loci of quantitative traits [14]. The genetic controls of many agronomic traits have been identified by GWAS [15,16,17,18,19,20,21,22,23,24]. For example, in recent years, several SNPs related to MC and DR were detected by GWAS at different stages of maize kernel development. Using an association panel containing 513 maize inbred lines, Li et al. [25] identified 71 SNPs affecting MC by GWAS. Combined with genetic population analysis, transcriptome analysis, and gene editing, two key genes *Zm00001d020929* and *Zm00001d046583* were shown to negatively regulate the MC. A total of 16 SNPs significantly associated with MC were detected by GWAS among a natural panel consisting of 310 maize inbred lines at the harvest stage [26]. Using GWAS, Li et al. [27] identified 27 SNPs related to kernel water content (KWC) and kernel dehydration rate (KDR) in maize. In addition, 334 SNPs were detected that significantly controlled the kernel moisture content (KMC) and KDR in 132 maize inbred lines before physiological maturity [28]. Despite the conduct of these studies, the genetic control of maize kernel and cob moisture changes at the physiological maturity stage remains unclear.

In this study, we measured cob moisture content (CMC), cob dehydration rate (CDR), KMC, and KDR values of 241 maize inbred lines on the 45th, 50th, and 55th days after pollination (DAP). GWAS was performed using the fixed and random circulating probability unification (FarmCPU) model to uncover the important SNPs and causal genes that control MC and DR. In addition, candidate gene association studies were conducted to elucidate the intragenic variations affecting the target traits. This study sought to (i) reveal the genetic basis of moisture change in maize kernel and cob, (ii) identify the hub genes affecting the maize moisture change, and (iii) contribute to the development of function markers for breeding low-MC and high-DR maize varieties through marker-assisted selection.

## 2. Results

### 2.1. Phenotypic Descriptions

The 12 traits investigated in the present study displayed great variation in the association panel of 241 maize inbred lines (Table 1). Most of the 12 traits followed normal distributions (Appendix A). At 45, 50, and 55 DAP, the mean values of CMC and KMC ranged from 57.01% to 62.60% and 26.98% to 35.70%, respectively (Table 1). The CMC and KMC both significantly declined from 45 to 55 DAP (Figure 1A). For DR at two successive time spans, the means of CDR and KDR were 0.95%–1.11% and 0.85%–1.16%, respectively (Table 1). The CDR and KDR variations both significantly (*p* < 0.05) increased from 45 to 55 DAP (Figure 1B). The coefficients of variation (CVs) of these12 traits were in the range of 0.61% to 10.68% (Table 1). In addition, the mean value of broad-sense heritability (*H*^2^) was 57.34% (Table 1), which indicated that the MC-related traits were mainly controlled by genetic factors.

### 2.2. SNPs and Candidate Genes Associated with Moisture Changes

GWAS was performed for MC and DR to explore the genetic basis of moisture changes. A total of four, three, and three significant (*p* < 1.82 × 10^−6^) SNP markers were detected for CDR45–55, CDR50–55, and KMC45, respectively, which distributed on chromosomes 1, 2, 3, 5, and 7 (Figure 2 and Appendix A). Among these significant SNPs, one (SYN38588) was simultaneously associated with CDR50–55 and CDR45–55 and had the lowest *p*-value (*p* = 9.58 × 10^−11^) in CDR45–55 associations (Appendix A). Based on our previous reports, the linkage disequilibrium (LD) decay was around 220 kb for this maize panel [29]. Thus, we searched the gene models in the LD regions of the significant SNPs. In total, we found 91 moisture-change-associated genes (Table 2 and Appendix A). For CDR50–55 and CDR45–55, one SNP, SYN38588, was repeatedly detected (Appendix A). Within the LD region of the co-detected marker (SYN38588), 15 gene models were uncovered, of which four were functionally annotated and the other 11 were unknown genes in maize (Table 2). Notably, the two genes *Zm00001d020615* and *Zm00001d020623* separately encode polygalacturonase (PG) and copper transporter 5.1 (COPT5.1), respectively. According to previous studies, PG and COPT5.1 proteins are involved in regulating the MC and DR of soybean and rice seeds [30,31,32,33,34]. To further understand the functions of the 11 genes, we performed a homology analysis and found six homologous genes in rice, sorghum, and *Panicum virgatum* (Appendix A). However, the encoding proteins of these homologous genes had not been reported to be associated with moisture changes. Therefore, *Zm00001d020615* and *Zm00001d020623* were selected as the prioritized candidate genes that controlled the moisture changes of maize at physiological maturity.

### 2.3. Candidate Gene Association Analysis Revealed Intragenic Variations Affecting Moisture Changes

To identify the causal gene(s) that control MC and DR at the physiological maturity stage in maize, we performed candidate gene association analyses for *Zm00001d020615* and *Zm00001d020623* using 67 randomly selected lines from the maize association panel. We detected 29 (27 SNPs and two indels) and nine (eight SNPs and one indel) polymorphic sites in the promoter (upstream 2000 bp), UTR, and gene body regions of *Zm00001d020615* and *Zm00001d020623*, respectively (Appendix A). Candidate gene association analyses showed that three significant SNPs (*p* < 0.05/19 = 2.63 × 10^−3^) in *Zm00001d020615* were identified, involving two SNPs (S7_125166053 and S7_125166495) related to KDR45–50, and one CDR50–55-associated SNP (S7_125165191) (Figure 3A, Table 3). However, no significant variation loci were identified for *Zm00001d020623*.

Among the three significant SNPs detected in *Zm00001d020615*, one missense variation (C/A) occurred in the fifth exon, and the other two synonymous variations (G/A, C/T) were located in the third and fifth exons, respectively (Figure 3A, Table 3). The 67 inbred lines were classified into three haplotypes (Hap1: AATTAA, Hap2: GGCCAA and Hap3: GGCCCC) based on the three variations. The lines with Hap1 had a significantly higher CDR50–55 value than those with Hap 3 (*p* < 0.05), whereas no significant difference in moisture-change-related phenotypes was observed between Hap2 and each of Hap1 and Hap3 (Figure 3B). As such, Hap1 was considered as the superior haplotype in this study. All these findings suggest that *Zm00001d020615* is a key gene affecting moisture changes of maize at the physiological maturity stage.

### 2.4. Distributions of Superior Alleles in Elite Lines

Among the association panel, 30 elite lines with excellent agronomic traits have been widely used as the parent lines for cultivating commercial maize varieties [29]. Evaluation of the utilization of superior alleles for the significant SNPs controlling the moisture-change-related traits is urgently needed for the breeding of varieties with low MC and high DR. Herein, the alleles that were associated with higher DR and lower MC values were defined as the superior alleles; conversely, the alleles associated with lower DR and higher MC values were designated as the inferior alleles. For each SNP, the ratio of superior to inferior alleles in the elite inbred lines was defined as the number of inbred lines containing superior alleles divided by the total number of inbred lines. Among the 30 elite inbred lines, the superior allele ratio for the nine significant SNPs ranged from 10% (SYN11290 and SYN30412) to 90% (PZE-102053209) (Figure 4). For seven SNPs associated with CDR traits, the superior allele ratios of three loci (PZE-103097076, PZE-102053209, and PZE-101122473) exceeded 50%, and the ratios of superior alleles for the other three loci (SYN38588, SYN11290, and SYN30412) were less than 20% (Figure 4). In addition, three SNPs (SYN15586, PZB01400.1, and SYN8680) associated with KMC all had low superior allele ratios, which were less than 20% (Figure 4). These results indicated that the superior alleles of CDR and KMC at physiological maturity were not widely used in maize breeding among the 30 elite inbred lines. Therefore, more superior alleles related to high DR and low MC need to be integrated into these elite inbred lines, which would be of practical significance for improving the dehydration characteristics of maize at physiological maturity.

## 3. Discussion

### 3.1. Dissecting the Genetic Basis of Moisture Changes at Physiological Maturity in Maize Using GWAS

Abundant phenotypic variation is an important factor for the successful analysis of the genetic basis of target traits. Population structure analysis showed that the association panel used in this study had high genetic diversity, including a tropical group, a stiff stalk (SS) group and a non-stiff stalk (NSS) group [35]. The MC and DR values ranged from 11.46% to 78.75% and 0.05% to 4.78% (with an average of 0.92%), respectively, suggesting that this panel was appropriate for GWAS. In addition, the mean value of *H*^2^ among 12 traits was 57.34%, which indicated that genetic factors were primary for these traits. Thus, it is feasible to decompose the genetic control of moisture changes at physiological maturity stage using a GWAS strategy. Moreover, a total of nine SNPs were associated with CDR45–50, CDR50–55, and KMC45, among which five SNPs were located within (or were close to) the kernel dehydration-associated QTL in maize reported in previous studies (Appendix A). For instance, SYN38588 and PZE-101122473 identified in this study were situated in the intervals of QTL (PZE-107014666/PZE-107082484 and PZE-101120411/PZE-101128157, respectively) that controlled dehydration initial moisture and initial time in maize kernels [11]. PZB01400.1 and SYN8680 were both contained within kernel dehydration initial moisture-associated QTL (PZE-101235343/SYN25670) [11]. The distance between the significant SNP SYN30412 identified in this study and the kernel final moisture-associated QTL (SYN30432/PZE-105040767) reported by Yin et al. was 0.08 Mb [11]. For the significant SNP PZB01400.1 identified in this study, a nearby (312 bp) significant SNP_286449826 associated with KMC at 40 DAP was detected by Li et al. [25]. These findings suggested that the genetic loci detected in this study were reliable. Notably, the lead SNP SYN38588 (*p* = 9.58 × 10^−11^) was repeatedly detected for traits CDR45–55 and CDR50–55. Based on this SNP, we identified two further priority genes controlling moisture changes. A candidate gene association study revealed that three significant SNPs located in the coding sequence of *Zm00001d020615* were associated with CDR50–55. In addition, we identified the favorable haplotype AATTAA, which should be given priority in marker-assisted selection breeding to cultivate maize varieties with low MC and high DR.

### 3.2. Candidate Genes Involved in MC and DR at Physiological Maturity Stage

At physiological maturity, the MC and DR of maize are mainly determined by two factors: stress dehydration caused by field environmental conditions and physiological dehydration caused by seed development and maturation [25]. Some specialized proteins that inhibit early embryo germination and avoid damages to embryo development play crucial roles in the process of grain physiological dehydration. In the present study, combining GWAS and candidate gene association analysis, we identified a key gene *Zm00001d020615*, which encodes a PG. Based on previous research, most of the PGs in plants belong to cell-wall localized pectin degrading enzymes and are expressed in different tissues and development stages [36]. PGs are involved in seed development, fruit softening, organ shedding, pollen ripening, and adversity stress [36]. To control water inflow and exit in soybean seeds, the PG encoding gene *PG031* affects seed coat permeability by regulating the intracellular space of parenchyma tissue and maintaining the integrity of the osteosclerosis layer [30]. In tomato, the PG gene *SlPG* has been confirmed as a key gene for improving fruit firmness. After mutagenesis of the *SlPG* gene, the hydrolysis of cell wall polymers decreased, which led to water loss reduction and delay in the softening of tomato fruit [37]. In rice, the *PSL1* gene functions as a PG that plays an important role in modifying cell wall biosynthesis of root and leaf and reducing water loss under drought stress [38].

### 3.3. Application of Superior Alleles in Breeding Maize Varieties with Low MC and High DR

In our study, SNP markers were provided for breeding new maize varieties with low MC and high DR at physiological maturity. By analyzing the utilization of superior alleles in 30 elite maize inbred lines, the superior allele ratios of three SNPs were found to be greater than 50%—one, PZE-102053209, was 90% (Figure 4). This indicated that the superior alleles of the three SNP were well maintained by artificial selection. A possible reason is that these alleles are closely linked with the agronomic traits of interest to breeders. Owing to natural variation in the association panel, researchers can develop molecular markers for breeding ideal maize varieties. In addition, only six lines contained more than five superior alleles (Figure 4), indicating that breeders have probably paid little attention to maize moisture changes at the physiological maturity stage. Therefore, the proportion of favorable alleles in 30 elite lines should be improved in cultivating maize varieties with low MC and high DR by marker-assisted selection breeding. The lines Mo17, Chuan273, Zheng22, and SCLM202, containing six superior alleles, are considered as excellent resources for breeding maize varieties with low MC and high DR through backcross breeding. In future research, the function of *Zm00001d020615* needs, firstly, to be verified. Then, gene *Zm00001d020615* can be overexpressed/knocked out to create maize lines with high DR.

## 4. Materials and Methods

### 4.1. Plant Materials

The association panel consisted of 241 diverse maize inbred lines (Appendix A), which were collected from the southwest China breeding program [35]. These lines were planted in greenhouses using a randomized complete block design with three replicates. The greenhouse parameters were set as follows: light/darkness = 16/8 h; temperature under light/temperature in darkness = 25/22 °C; relative humidity = 65%. In each replicate, every line was grown in one row with 14 plants. The distance between two rows was 0.7 m, and the row length was 3 m. To ensure the consistency of sampling, we selected the plants that grew uniformly for investigating the target traits.

### 4.2. Phenotypic Collection and Data Analysis

For each line, the fresh weights (FWs) of the cob and kernel were measured separately on the 45th, 50th, and 55th DAP. Then, the dry weights (DWs) of the cob and kernel were measured after drying at 80 °C in an oven for 72 h. Three individuals were weighed for every line at each stage. The MC was calculated as follows: moisture content (%) = (FW − DW)/FW × 100%. According to the above method, cob moisture content (CMC) and kernel moisture content (KMC) were calculated on the 45th, 50th, and 55th DAP, respectively. Based on the MC values at two successive stages, DR was calculated as follows: dehydration rate (45–50 d) (%) = (MC on 45 d-MC on 50 d)/5 × 100%; dehydration rate (50–55 d) (%) = (MC on 50 d-MC on 55 d)/5 × 100%; dehydration rate (45–55 d) (%) = (MC on 45 d-MC on 55 d)/10 × 100%. The mean value across three replicates was used as the final phenotype value. The KMCs on 45, 50, and 55 DAP were designated as KMC45, KMC50, and KMC55. The CMCs on 45, 50, and 55 DAP were designated as CMC45, CMC50, and CMC55. The kernel DRs for the three time spans (namely, 45–50, 50–55, and 45–55 DAP) were denoted as KDR45–50, KDR50–55, and KDR45–55. The cob DRs for the three time spans (namely, 45–50, 50–55, and 45–55 DAP) were denoted as CDR45–50, CDR50–55, and CDR45–55. Descriptive statistical analysis was performed using SPSS25 software. The broad-sense heritability (*H*^2^) was calculated as follows [39]: *H*^2^ = σ_G_^2^/σ_P_^2^, σ_G_^2^ = (MSG-MSE)/rep, σ_P_^2^ = (MSG-MSE)/rep + MSE. Herein, σ_G_^2^, σ_P_^2^, MSG, MSE, and rep represent genotypic variance, phenotypic variance, mean square of genotype, mean square of error, and number of replicates, respectively.

### 4.3. Genome-Wide Association Study

In our previous study, genotyping of the maize panel was performed by Illumina Maize SNP50K Bead Chip, and a total of 56,110 SNPs were detected [35]. According to filtration criteria of missing rate >20% and minor allele frequency (MAF) ≤ 0.05, 46,603 SNPs were retained for GWAS. Cluster analysis revealed that the 241 inbred lines could be divided into three types: a tropical group (population 1), a non-stiff stalk (NSS) group (population 2), and a stiff stalk (SS) group (population 3) based on the 46,603 SNPs (Appendix A). As the FarmCPU model was superior to other models (general linear model and mixed linear model) in balancing false positives/negatives [40], we used the FarmCPU to detect the associations for moisture changes. The model was executed by the R Studio ver.4.0.3 (Allaire, Boston, MA, USA) with a FarmCPU package [41]. The simpleM program in R Studio ver.4.0.3 (Allaire, Boston, MA, USA) [42] was applied to calculate the effective marker number of independent tests (M_eff_G_ = 27,507) [42,43,44]. The significance threshold was calculated as *p* = 0.05/M_eff_G_ = 1.82 × 10^−6^ [26]. In addition, the gene models that were located in the linkage disequilibrium (LD) region of each SNP were considered the candidate genes. Combined with the gene functional annotations from NCBI (NCBI, National Center for Biotechnology Information, https://www.ncbi.nlm.nih.gov/, accessed on 24 February 2022) databases, we determined the hub genes which were related to moisture changes.

### 4.4. Candidate Gene Association Study

We amplified the hub gene sequences, including the promoter (upstream 2000 bp), UTR, and gene body regions in 67 inbred lines, randomly selected from three subgroups of the association panel (Appendix A). The PCR-amplified sequences were aligned with the B73 (v4) genome using DNAMAN ver.5.2.2 (Reachsoft, Beijing, China) [45]. The variations (SNPs and InDels) with MAF ≥ 5%, and the trait phenotypes, were used as the inputs for a candidate gene association study based on a general linear model in TASSEL ver.5.0 (Buckler lab, Cornell university, New York, NY, USA) software [23]. The significant threshold was set as: *p* = 0.05/n (n represents the number of SNPs). The LD decay between pairwise SNPs was calculated using Haploview.JRE ver.18.0.2 (The Broad Institute of MIT and Harvard, Cambridge, MA, USA) software [46]. Phenotypic differences between haplotypes were analyzed using a *t*-test.

### 4.5. Analysis of Superior Alleles

In our study, alleles associated with low MC and high DR were considered as superior alleles. The ratio of superior alleles for each significant SNP was calculated as the number of lines with superior alleles divided by the total number of lines [19]. According to the rate of superior alleles in each line, we generated a visualized heat map using the heatmap package in R Studio ver.4.0.3 (Allaire, Boston, MA, USA) software.

## Figures and Tables

**Figure 1 plants-11-01989-f001:**
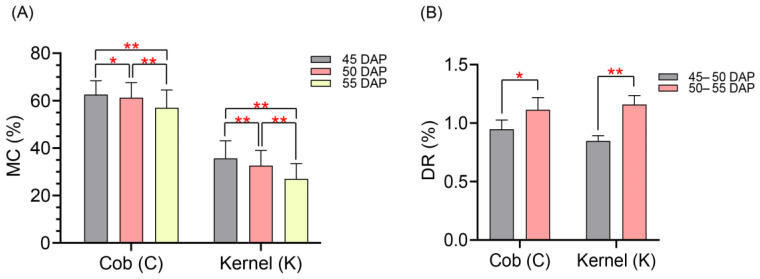
Phenotypes of moisture-content-related traits of the association panel at physiological maturity stages. (**A**) Phenotypic values of CMC and KMC measured at each stage. MC, moisture content. (**B**) Phenotypic values of CDR and KDR measured for the two time spans. DR, dehydration rate. * Significant at *p* < 0.05. ** Significant at *p* < 0.01.

**Figure 2 plants-11-01989-f002:**
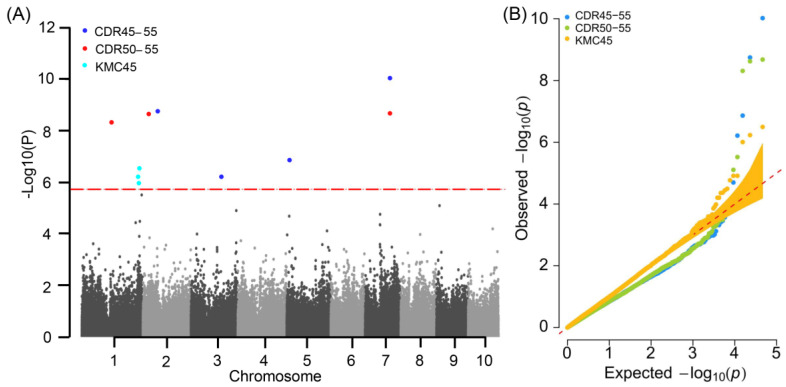
Significant SNPs detected by GWAS using FarmCPU model. (**A**) Manhattan diagram of GWAS results for moisture changes. The blue, red, and turquoise dots represent the significant SNPs associated with CDR50–55, CDR45–55, and KMC45, respectively. CDR50–55 and CDR45–55 represent cob dehydration rate at 50–55 and 45–55 days after pollination, respectively; KMC45 represents kernel moisture content on the 45th day after pollination. The red broken line represents the significant threshold of 1.82 × 10^−6^. (**B**) Quantile–quantile (Q–Q) plot of GWAS results for moisture changes. The blue, green, and yellow dots represent the SNPs associated with CDR50–55, CDR45–55, and KMC45, respectively. CDR50–55 and CDR45–55 represent cob dehydration rate at 50–55 and 45–55 days after pollination, respectively. KMC45 represents kernel moisture content on the 45th day after pollination.

**Figure 3 plants-11-01989-f003:**
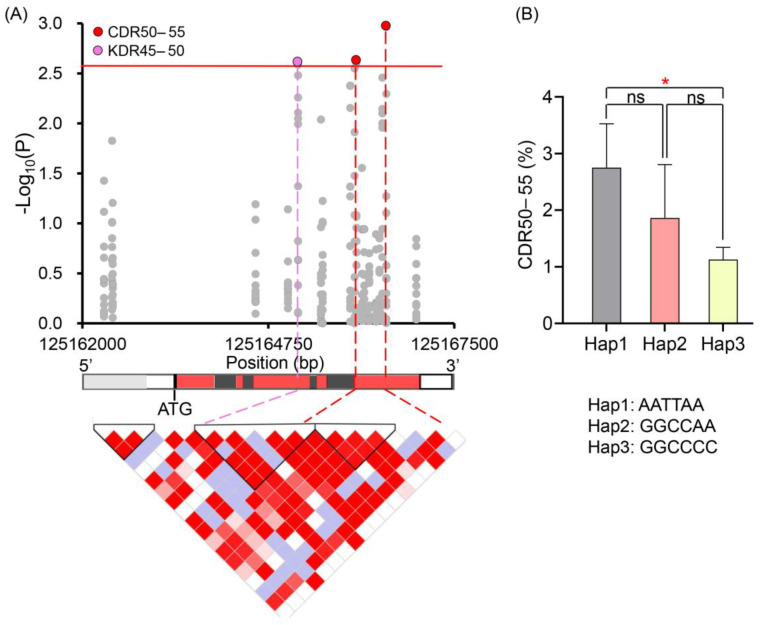
Association analysis of *Zm00001d020615*. (**A**) Significant SNPs associated with CDR50–55 and KDR45–50. CDR50–55 represent cob dehydration rate at 50–55 days after pollination; KDR45–50 represent kernel dehydration rate at 45–50 days after pollination. Red line shows the significant threshold of markers. The boxes with gray, white, red, and black colors represent promoter (upstream 2000 bp), UTR, exons, and introns, respectively. Bottom plot shows the pairwise linkage disequilibrium between the target SNP markers. (**B**) Comparison of three haplotypes for CDR50–55. CDR50–55 represent cob dehydration rate at 50–55 days after pollination. Hap1, haplotype1; Hap2, haplotype2; Hap3, haplotype3. * Significant at *p* < 0.05. ns, no significant.

**Figure 4 plants-11-01989-f004:**
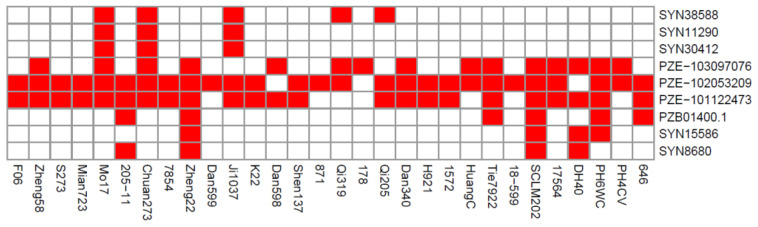
Superior allele distributions of nine SNPs in 30 elite inbred lines. Red and white colors represent superior and inferior alleles, respectively.

**Table 1 plants-11-01989-t001:** Phenotypic variations of 12 traits in 241 maize inbred lines.

Trait	Mean (%)	Max (%)	Min (%)	SD	CV (%)	*H*^2^ (%)
CMC45	62.60	78.75	46.37	0.38	0.61	62.38
CMC50	61.26	77.28	42.30	0.48	0.79	73.69
CMC55	57.01	71.80	35.52	0.71	1.25	65.25
KMC45	35.70	73.50	22.52	0.49	1.37	29.79
KMC50	32.59	65.55	14.46	0.49	1.50	35.15
KMC55	26.98	46.72	11.46	0.61	2.26	68.71
CDR45–50	0.95	4.78	0.01	0.08	8.15	92.16
CDR50–55	1.11	4.53	0.02	0.10	9.35	88.41
CDR45–55	0.72	3.41	0.01	0.08	10.68	60.41
KDR45–50	0.85	1.84	0.06	0.04	7.51	22.20
KDR50–55	1.16	2.56	0.05	0.08	8.75	24.52
KDR45–55	0.75	1.36	0.06	0.04	6.33	65.41

Max, maximum; Min, minimum; SD, standard deviation; CV, coefficient of variation; *H^2^*, broad-sense heritability. CMC45, CMC50, and CMC55 represent cob moisture content on the 45th, 50th, and 55th days after pollination, respectively; KMC45, KMC50, and KMC55 represent kernel moisture content on the 45th, 50th, and 55th days after pollination, respectively; CDR45–50, CDR50–55, and CDR45–55 represent cob dehydration rate at 45–50, 50–55, and 45–55 days after pollination, respectively; KDR45–50, KDR50–55, and KDR45–55 represent kernel dehydration rate at 45–50, 50–55, and 45–55 days after pollination, respectively.

**Table 2 plants-11-01989-t002:** Annotations of candidate genes scanned from co-localized significant SNP (SYN38588) detected by GWAS.

Associated Traits	Candidate Genes	Annotations
	*Zm00001d020618*	-
	*Zm00001d020610*	-
	*Zm00001d020609*	-
	*Zm00001d020612*	-
	*Zm00001d020622*	-
	*Zm00001d020615*	polygalacturonase
CDR45–55	*Zm00001d020626*	-
CDR50–55	*Zm00001d020613*	-
	*Zm00001d020617*	exocyst complex component EXO70B1
	*Zm00001d020614*	-
	*Zm00001d020623*	copper transporter 5.1
	*Zm00001d020616*	-
	*Zm00001d020627*	-
	*Zm00001d020628*	trimethyltridecatetraene synthase
	*Zm00001d020620*	-

CDR45–55 and CDR50–55 represent cob dehydration rate at 45–55 and 50–55 days after pollination, respectively. ‘-’ represents no functional annotations.

**Table 3 plants-11-01989-t003:** Gene-based association studies detected significantly associated variants within *Zm00001d020615*.

Trait	Marker	Position	*p*-Value	Allele	Variation Region
CDR50–55	S7_125165191	125165191	0.00259	G/A	the third exon (synonymous)
KDR45–50	S7_125166053	125166053	0.00246	C/T	the fifth exon (synonymous)
	S7_125166495	125166495	0.00108	C/A	the fifth exon (missense)

CDR50–55 represent cob dehydration rate at 50–55 days after pollination; KDR45–50 represent kernel dehydration rate at 45–50 days after pollination.

## Data Availability

The genotype data used in this study were described in a previous study. All datasets are available from the corresponding author on reasonable request.

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
