# Peer review of "Genome-Wide Association Study Reveals the Genetic Basis of Kernel and Cob Moisture Changes in Maize at Physiological Maturity Stage"

_plants, 2022, doi:10.3390/plants11151989_

Round 1

Reviewer 1 Report

In the manuscript entitled “Genome-wide association study reveals the genetic basis of kernel and cob moisture changes in maize at physiological maturity stage”, the authors have used an association panel composed of 241 maize inbred lines to discover the genetic loci and the causal genes underlying ear moisture changes at physiological maturity stage. The identified QTL and (or) candidate genes will be helpful to breed maize cultivars with high dehydration rates that were suitable for mechanical harvest. This manuscript will be of great interest of the readers in Plants, especially in crop sciences, thus it should be published in Plants.

Minor:

(1) Line 42-43: GRMZM2G137211 and GRMZM5G805627 may be updated according to AGPv4 with a prefix “Zm00001d”. This will be uniform because the authors have used gene models according to AGPv4 in the abstract.

(2) Line 62: “The most” is better than “Most”.

(3) Line 80: “content-related traits” is better than “content-related trait”.

(4) Line 88: “p” should be in the capital form.

(5) Line 89: please indicate “LD”.

(6) Line 113: in Table 2, the first column can be removed because this information has been indicated in the table title.

(7) Line 169: in Figure 4, the color bar at the right of the panel may be useless.

Reviewer 2 Report

The introduction is too short. Are there any QTL studies done using a bi-parental population that you can mention here?  

Line 38- Rewrite the statement “ GWAS has been widely used to identify single nucleotide polymorphisms (SNPs) for different maize”

Line 223 – The number “241” is different than your previous publication. Can you provide the list of lines used in this study?

Figure 1 – I can’t tell “Variations of moisture content-related trait among 241 inbred lines” based on the figure. Please change the title accordingly.

Figure 2- Can you change it to a rectangular Manhattan plot and add a QQ-plot on the side?

Reviewer 3 Report

Minyan Zhang et al. have send this MS ‘Genome-wide association study reveals the genetic basis of kernel and cob moisture changes in maize at physiological maturity stage’ to get published in Plants. They studied panel composed of 241 maize inbred lines to analyzed ear moisture changes at physiological maturity stage using GWAS. Their results These results revealed the genetic basis of ear moisture changes in maize at physiological maturity stage, and provided the superior haplotype for cultivating maize varieties with low moisture contents and high dehydration rates.

I found the MS is fit well to be published after the little comments be taken in consideration to improve the MS.

-          In line 13. One SNP SYN38588 was repeatedly….. please change to One SNP (SYN38588) was repeatedly….and in Table 2’ title also

-          This study aims to (1)……. the numbering could change to (i) and (ii) and so on… to avoid misleading with reference numbers

-          In addition, the mean value of broad-sense heritability (H2) was 57.34%..... where this result could found? Please refer to the position in S tables or data.

-          In Fig 1, please add the abbreviations, Cob (C), Kernel (K), MC and DR….

-          Lines 84 and 85; A total of four, three, and three significant SNP markers distributed on chromosomes 1, 2, 3, 5, and 7 were detected, How is it? What about Chr 5 and 7, something missing I think.

-          In line 96 and also in line 16, please add (…..transporter 5.1 (COPT5.1), respectively.

-          In line 97, ….. regulating MC and DR of seeds (of what, maiz or also other crops?)

-          ‘All these findings suggested that Zm00001d020615 was a key gene affecting moisture changes of maize at the physiological maturity stage’ this conclusion could improve the abstract, please add this result to abstract

-          In the discussion part 3.1 and 3.3, need to improve and discuss more not to repeat the results

Reviewer 4 Report

Zhang et al. performed a GWAS analysis on 241 maize inbred lines to unveil the genetic basis of kernel and cob moisture changes in maize at the physiological maturity stage. Overall, the study is interesting and could provide some insights into breeding. However, there are some concerns regarding the study and the authors may need to address them before the acceptance of this manuscript.

Major:

1.The authors may consider adding a population structure, PCA and phylogenetic analyses to show the clustering of the selected 241 maize inbred lines

2.During the selection of the GWAS model, the authors may need to explain a bit why FarmCPU was used since there are different models.

3.During GWAS, since KMC and KDR have been studied by others, why did the authors include them? any new findings compared to previous studies?

4.During candidate gene association analysis, 67 lines were selected. Here, the authors may need to explain the reasons to select 67 lines? Can the authors indicate those selected 67 lines in the phylogenetic tree requested in Q1?

5.L94: ’11 were unknown genes’ – Did the author perform any homologous comparison to identify the gene functions? or did the author use BLAST against the NCNI-nr database to identify the function? 

Minor:

1.The authors mentioned several times of ‘genetic factors’, what are they? Can the authors prvodie some examples?

2.L39 ‘In recent years’: may change to ‘For example, in recent years’

3.L90 ’91 gene models …’: does that mean all 91 gene models that are in the LD region have associations with moisture changes? Please reword to avoid misunderstanding

4.L134 ‘As such, the Hap1 was confirmed’: Is this true? What else did the authors perform to confirm this? If there are no other validations, please change ‘confirmed’ to another word to avoid misleading.

5.Section 3.3: The authors may discuss what kind of methods or strategies that could be used to help breeding

Round 2

Reviewer 4 Report

Thanks to the authors for the amendment. I don't have other questions.